# Immunotherapy and Advanced Vulvar Cancer: A Systematic Review and Meta-Analysis of Survival and Safety Outcomes

**DOI:** 10.3390/cancers17142392

**Published:** 2025-07-19

**Authors:** Mauro Francesco Pio Maiorano, Vera Loizzi, Gennaro Cormio, Brigida Anna Maiorano

**Affiliations:** 1Unit of Obstetrics and Gynecology, Department of Interdisciplinary Medicine (DIM), University of Bari “Aldo Moro”, Polyclinic of Bari, Piazza Giulio Cesare 11, 70124 Bari, Italy; m.maiorano23@studenti.uniba.it (M.F.P.M.); gennaro.cormio@uniba.it (G.C.); 2Unit of Oncologic Gynecology, IRCCS “Giovanni Paolo II” Oncologic Institute, Viale Orazio Flacco 65, 70124 Bari, Italy; vera.loizzi@uniba.it; 3Translational Biomedicine and Neuroscience Department (DiBraiN), University of Bari “Aldo Moro”, Piazza Giulio Cesare 11, 70124 Bari, Italy; 4Department of Medical Oncology, IRCCS San Raffaele Hospital, Via Olgettina 60, 20132 Milan, Italy

**Keywords:** immunotherapy for vulvar cancer, vulvar cancer staging, vulvar cancer treatment, immune checkpoint inhibitors vulvar cancer, vulvar cancer survival, vulvar cancer immunotherapy systematic review, immunotherapy vulvar cancer meta-analysis, PD-L1 immunotherapy vulvar cancer, vulvar cancer cohort study, efficacy of immunotherapy vulvar cancer

## Abstract

Advanced and recurrent vulvar squamous cell carcinoma remains difficult to treat, with limited therapeutic options and poor survival outcomes. Immune checkpoint inhibitors have shown promise in other HPV-associated cancers, but their role in VSCC is less defined. This systematic review and meta-analysis synthesized the current evidence on ICIs in advanced VSCC, demonstrating encouraging response rates in this challenging setting, with acceptable toxicity. Our findings highlight key limitations in biomarker reporting and underscore the need for more precise patient selection. This work provides a foundation for future clinical trials and biomarker-driven strategies aimed at improving outcomes in this rare and aggressive disease.

## 1. Introduction

Vulvar squamous cell carcinoma (VSCC) is a rare malignancy, representing approximately 4–5% of all gynecologic cancers worldwide [1]. In 2020, there were 45,240 new cases and 17,427 related deaths globally [2]. The incidence of VSCC varies by age, with younger patients often presenting with human papillomavirus (HPV)-associated tumors, whereas HPV-independent VSCC is more frequently observed in postmenopausal women and is usually linked to chronic inflammatory conditions such as lichen sclerosus [3,4]. While early-stage VSCC is managed primarily through surgical resection, with adjuvant radiotherapy or chemotherapy when indicated, advanced and recurrent disease presents a significant therapeutic challenge [5]. Patients with locoregional recurrence have a five-year survival rate of approximately 50–70%, which decreases to 27% in cases of lymphatic recurrence and 14% in cases of distant metastatic disease [6]. Given these poor outcomes, novel therapeutic strategies are urgently needed. The immune system plays a critical role in tumor surveillance and response, with immune evasion mechanisms contributing to tumor progression [7,8]. Immune checkpoint inhibitors (ICIs) have revolutionized cancer therapy by targeting inhibitory pathways such as the programmed death (PD)-1/PD-ligand (L)1 axis, which tumors exploit to evade immune detection [9]. Several biomarkers, including PD-L1 expression, microsatellite instability (MSI)/deficient mismatch repair (dMMR), tumor mutational burden (TMB), and the presence of tumor-infiltrating lymphocytes (TILs), have been identified as potential predictors of response to ICIs in other solid tumors [9,10,11,12]. While pembrolizumab and other ICIs have demonstrated efficacy in HPV-related malignancies, such as cervical and head and neck cancers, their role in VSCC remains less well-defined due to the rarity of the disease and the limited availability of large prospective trials [13,14,15,16]. In fact, VSCC exhibits several immunological features that support the rationale for the use of ICIs. The tumor immune microenvironment is often rich in immune cell infiltrates, particularly in HPV-positive VSCC, where the presence of CD4^+^ and CD8^+^ T cells has been associated with better prognosis [17]. However, VSCC frequently evades immune responses through mechanisms such as PD-L1 overexpression, indoleamine 2,3-dioxygenase (IDO) production, and transforming growth factor-beta (TGF-β) signaling, all of which contribute to T-cell suppression and tumor progression [18,19,20,21,22,23]. Importantly, PD-L1 is expressed in up to 40% of VSCC cases, especially in HPV-negative tumors, and this expression correlates with worse outcomes [23]. These findings suggest that blocking the PD-1/PD-L1 axis in VSCC could restore anti-tumor immunity. Clinical trials support this approach: in the KEYNOTE-028 trial, pembrolizumab showed a partial response in one of two patients with PD-L1-positive VSCC, and in the CheckMate 358 trial, nivolumab resulted in disease stabilization in three of five patients [24,25]. Taken together, the immune characteristics of VSCC and early clinical signals suggest that ICIs may represent a promising therapeutic option, particularly in selected patients. In fact, while the U.S. Food and Drug Administration (FDA) has approved pembrolizumab for any solid tumor with PD-L1 positivity, TMB-high, or MSI-high/dMMR status [26], the National Comprehensive Cancer Network (NCCN) guidelines recognize pembrolizumab as a second-line option for advanced or metastatic VSCC, meeting the same biomarker criteria [27]. However, the European Medicines Agency (EMA) has yet to approve ICIs for VSCC, highlighting the ongoing need for further clinical research. Given this increasing interest in immunotherapy for gynecologic malignancies and the lack of high-level evidence that may serve as a basis for clearer clinical guidelines, this study aims to systematically review and meta-analyze the current evidence on the efficacy and safety of ICIs in advanced VSCC, summarizing the available clinical trial data and exploring potential avenues for future therapeutic development.

## 2. Materials and Methods

The systematic review and the meta-analysis were conducted following the recommendations of the Cochrane Systematic Reviews, and our findings are reported according to the Preferred Reporting Items for Systematic Reviews and Meta-Analyses (PRISMA) guidelines [28]. We prospectively registered our systematic review on the international database of prospectively registered systematic reviews, PROSPERO (registration ID: CRD420251067565). Ethical approval was not required, due to the inclusion of anonymized, previously published data.

### 2.1. Search Strategy

A comprehensive literature search was conducted to identify studies evaluating the role of ICIs in patients with advanced (unresectable, recurrent, or metastatic) VSCC. Two authors (MFPM and BAM) independently conducted the research, and discrepancies were resolved through consensus. The databases searched included PubMed, Embase, Scopus, and the Cochrane Library, covering studies published in the last ten years, from 1 January 2015 to 31 May 2025. The search strategy combined medical subject headings (MeSH) and free-text terms such as “vulvar cancer”, “vulvar squamous cell carcinoma”, “immune checkpoint inhibitors”, “pembrolizumab”, “nivolumab”, “cemiplimab”, “ipilimumab”, “immunotherapy”, “HPV”, “PD-1”, “CTLA-4”, and “PD-L1”, using Boolean operators (AND/OR) to maximize sensitivity. Additionally, reference lists of systematic reviews and included studies were manually screened to identify additional eligible articles that may not have been captured in the initial database search. The full search strategy is available (Appendix A).

### 2.2. Eligibility Criteria

Studies were included if they met predefined eligibility criteria based on the Population, Intervention, Comparison, Outcomes, and Study design (PICOS) framework (Appendix A) [29]. Eligible studies enrolled patients with advanced, unresectable, recurrent, or metastatic VSCC who received treatment with ICIs, including anti-PD-1, anti-PD-L1, or anti-cytotoxic T-lymphocyte antigen (CTLA)-4 monoclonal antibodies, either as a monotherapy or in combination with other agents, chemo, or radiotherapy. No restrictions were applied regarding previous therapies, but patients had to be immunotherapy-naive. Also, no restrictions were applied regarding comparator arms, allowing the inclusion of randomized controlled trials (RCTs) and single-arm clinical trials. To ensure clinical relevance, only full-text studies reporting at least one of the following outcomes were included: objective response rate (ORR), progression-free survival (PFS), overall survival (OS), or safety data. Studies were excluded if they were written in a language other than English, were not published in a peer-reviewed journal, or were focused on non-immunotherapeutic agents, preclinical or translational research, or if they were reviews, commentaries, meta-analyses, cohort studies, case series, case reports, editorials, or conference abstracts without extractable patient-level clinical data.

### 2.3. Study Selection and Data Extraction

The study selection process adhered to the PRISMA guidelines, ensuring transparency and reproducibility [28]. The PRISMA flow diagram (Figure 1) outlines the number of records identified, screened, and included, along with the exclusion reasons. Two independent reviewers (MFPM and BAM) screened the titles and abstracts of all retrieved studies, followed by a full-text review of potentially eligible articles. Any discrepancies in study selection were resolved by consensus or consultation with a third reviewer (VL).

### 2.4. Data Extraction

Data extraction was performed systematically using a standardized template. The extracted variables included study design (e.g., phase of the trial, single-arm vs. randomized), sample size, patient characteristics such as age, stage at diagnosis, PD-L1, HPV status, treatment regimen including drug type, dosage, and administration schedule, and clinical outcomes. The efficacy endpoints included ORR, PFS, and OS, with corresponding 95% confidence intervals (CI), while the safety endpoints included the incidence of treatment-related adverse events (AEs) categorized by severity (Grade 1–5). When needed, corresponding authors were contacted for additional data.

### 2.5. Risk of Bias Assessment

Risk of bias was independently assessed by two reviewers (MFPM and BAM) using the Methodological Index for Non-Randomized Studies (MINORS) tool, which includes eight domains: clearly stated aim, inclusion of consecutive patients, prospective data collection, appropriate endpoints, unbiased assessment of the study endpoint, and adequate follow-up period [30]. Each item was scored from 0 (not reported) to 2 (reported and adequate), with discrepancies resolved through discussion and consensus. This assessment aimed to evaluate the internal validity of the included single-arm trials and identify potential sources of bias related to the study design, data collection, and reporting (Appendix A).

### 2.6. Data Synthesis and Statistical Analysis

Descriptive statistics were used to summarize the characteristics of the included studies. A meta-analysis was performed to derive pooled estimates for ORR, PFS, and OS using a random-effects model (DerSimonian and Laird method) to account for inter-study heterogeneity [31,32]. The degree of heterogeneity across studies was assessed using Cochran’s Q and I^2^ statistic, with an I^2^ value exceeding 50% being indicative of substantial heterogeneity [33]. Pooled estimates for ORR were calculated with a 95% CI, and time-to-event analyses for PFS and OS were pooled where Kaplan–Meier estimates were available. Forest plots were generated to visualize study-specific and pooled ORR, PFS, and OS estimates. Publication bias was assessed using funnel plots when the number of studies permitted meaningful interpretation. All statistical analyses were conducted using SPSS version 24 [34] and R, meta package version 4.4.2 [35]. Subgroup analyses were conducted for ORR based on PD-L1 expression. The overall certainty of evidence was assessed using the GRADE approach [36] (Appendix A).

## 3. Results

A total of 41 records were identified through a systematic search of PubMed, Embase, Scopus, and the Cochrane Library. After the removal of 7 duplicates, 34 studies remained for title and abstract screening. Of these, 21 were excluded for being reviews, editorials, preclinical studies, or unrelated to the analyzed topic. Thirteen full-text articles were assessed for eligibility, and seven were excluded for not reporting extractable efficacy or safety data on ICIs in VSCC. Ultimately, six studies met the predefined inclusion criteria and were included in the qualitative and quantitative synthesis (Figure 1).

### 3.1. Characteristics of the Included Studies and Main Overall Findings

This systematic review includes six non-randomized, single-arm clinical trials investigating ICIs in advanced, unresectable, recurrent, or metastatic VSCC [24,25,37,38,39,40]. These studies evaluated pembrolizumab, nivolumab, and ipilimumab as a monotherapy or in combination with other agents, chemo, and/or radiotherapy. The trials included were four phase II trials (KEYNOTE-158, PEVOsq, Yeku et al., SWOG S1609), one phase Ib trial (KEYNOTE-028), and one phase I/II trial (CheckMate 358) [24,25,37,38,39,40]. Across the nine studies included in this systematic review, the ORR ranged from 5.6% to 75%, the median progression-free survival (mPFS) ranged from 1.3 months to 2.5 months, and the median overall survival (mOS) varied from 3.7 to 17.5 months. Safety outcomes demonstrated manageable toxicity profiles, with treatment-related adverse events (AEs) occurring in 50.5% to 100% of patients. In 0% to 78.6% of cases, ≥G3 AEs were reported, leading to treatment discontinuation in up to 4% of patients, and a total of three treatment-related deaths (1.66%) were reported. Table 1 synthesizes the main characteristics and findings from the included studies.

### 3.2. Patient Population, Previous Therapies, and Intervention

A total of 181 patients with advanced, unresectable, metastatic, or relapsed VSCC were included across all studies. The median age of participants across the included studies ranged from 55.5 to 64 years. In the KEYNOTE-158 trial, 101 patients with previously treated advanced VSCC were enrolled. Median follow-up from the first dose was 36.0 months (range, 15.4–55.2). The median age was 64 years (range, 31–87), and most patients had an ECOG performance status of one (84.2%) and stage IV disease (87.1%). Notably, 33.7% had received two or more prior lines of systemic therapy, and 92.1% had previously undergone radiation therapy. PD-L1 expression was assessed using the PD-L1 IHC 22C3 pharmDx assay, with PD-L1 positivity defined as a combined positive score (CPS) ≥ 1, and 83.2% of patients had PD-L1-positive tumors. The patients were not tested for HPV. Pembrolizumab was administered at a dose of 200 mg intravenously every 3 weeks, continued for up to 35 cycles or until disease progression, unacceptable toxicity, or other protocol-defined discontinuation criteria [37]. In KEYNOTE-028, 18 patients with PD-L1 positive pre-treated with chemotherapy and/or radiotherapy VSCC were included. PD-L1 positivity was assessed using the Dako 22C3 assay with a CPS ≥ 1 cutoff. No HPV data were reported. The treatment consisted of pembrolizumab 10 mg/kg IV every 2 weeks and continued for up to 2 years or until progression [24]. The phase II PEVOsq trial enrolled 17 patients with advanced VSCC, including locally recurrent and metastatic disease, with a median age of 63 years (range 40–85). These patients received a first-line treatment with pembrolizumab (200 mg IV every 3 weeks) in combination with the histone deacetylase inhibitor (HDAC) vorinostat (400 mg orally daily) and were not selected based on HPV or PD-L1 status [38]. In the single-arm phase II trial conducted by Yeku et al. (NCT04430699), 24 patients with advanced VSCC were enrolled (92% [n = 22] had primary unresectable disease, and 8% [n = 2] presented with recurrent disease) and treated with curative-intent chemoradiotherapy combined with immunotherapy. The eligible patients included those with prior chemotherapy exposure. The treatment protocol consisted of weekly cisplatin (40 mg/m^2^) administered concurrently with intensity-modulated radiotherapy (IMRT), alongside pembrolizumab 200 mg given intravenously every three weeks for a total of 12 cycles. All participants received definitive radiotherapy, with a median dose of 68.4 Gy to the primary tumor (range, 26.2–70.2 Gy) and 45 Gy to the pelvic, inguinal, and vulvar clinical target volumes (range, 21.6–50.4 Gy [39]). The CheckMate 358 trial included five patients with (2 HPV-positive and 3 HPV-negative) PD-L1-positive VSCC, of whom four had previously received systemic platinum-based chemotherapy. PD-L1 expression was assessed using the Dako 28-8 assay, with a CPS ≥ 1 cutoff. The treatment regimen was nivolumab 240 mg IV every 2 weeks, continued for up to 2 years or until progression [25]. The SWOG S1609 Dual Anti–CTLA–4 and Anti–PD–1 Blockade in Rare Tumors (DART) trial is a multicenter, prospective, open-label phase II study evaluating dual immune checkpoint inhibition with ipilimumab and nivolumab in rare cancers, including VSCC. This multi-cohort trial included a vulvar cancer cohort in which patients received nivolumab 240 mg intravenously every 2 weeks and ipilimumab 1 mg/kg every 6 weeks. Sixteen evaluable patients with VSCC, all with squamous histology and a median age of 55.5 years, were included in the analysis. None had received prior immunotherapy, and the number of prior systemic therapies ranged from 0 to 6. No HPV or PD-L1 data were reported for this population [40].

### 3.3. Efficacy Outcomes

In the KEYNOTE-158 trial, the ORR in the total population was 10.9% (95% CI, 5.6–18.7%), including one complete response (CR) and ten partial responses (PRs). Clinical benefit—defined as CR, PR, or stable disease (SD)—was observed in 28.7% of patients. Among the 78 patients with at least one post-baseline tumor assessment, 43.6% experienced a reduction in target lesion size compared to baseline. The median time to response among those achieving CR or PR was 4.0 months (range, 1.6–6.0), and the median duration of response (DOR) was 20.4 months (range, 2.1+ to 28.0), with 80% of responders maintaining response for ≥6 months and 60% for ≥12 months. Responses were sometimes sustained after the discontinuation of pembrolizumab. The ORR was 9.5% (95% CI, 4.2–17.9%) among PD-L1-positive tumors (CPS ≥ 1) and 28.6% (95% CI, 3.7–71.0%) among PD-L1-negative tumors (CPS < 1). Objective responses occurred in 7.5% of patients who had received 0–1 prior lines of therapy and in 17.7% of those with ≥2 prior lines. At data cutoff, 94.1% of patients had experienced disease progression or death. The median progression-free survival (mPFS) was 2.1 months (95% CI, 2.0–2.1), and the median overall survival (mOS) was 6.2 months (95% CI, 4.9–9.4). Estimated 6- and 12-month PFS rates were 19.8% and 9.9%, respectively, while 6- and 12-month OS rates were 50.5% and 34.7% [37]. In KEYNOTE-028, the ORR was 6%, with an mPFS of 3.8 months (95% CI: 1.4–6.2) and an mOS of 3.8 months (95% CI: 2.6–5.9) [24]. The PEVOsq trial demonstrated an ORR of 18.8% (1 CR, 2 PRs). The mPFS was 1.3 months (95% CI: 1.1–4.3), and the mOS was 17.5 months (95% CI: 2.3–not reached) [38]. In the trial by Yeku et al., the ORR was 75%. Among the 24 enrolled patients, one discontinued RT early due to progressive disease (PD). The six-month recurrence-free survival (RFS) rate was 70% (95% CI, 48–85%). At the time of analysis, the mPFS had not yet been reached, indicating sustained disease control in a significant proportion of patients [39]. The CheckMate 358 trial reported an ORR of 20% (95% CI: 0.5–71.6%), with a 6-month PFS rate of 40%, a 12-month OS rate of 40%, and an 18-month OS rate of 20% [25]. In the vulvar cancer cohort of the SWOG S1609 DART trial, the ORR was 18.8% (3/16), including one CR and two confirmed PRs. The clinical benefit rate (CBR), defined as ORR plus stable disease lasting at least six months, was 25% (4/16), and increased to 31% (5/16) when including one unconfirmed PR. The PFS durations for these five patients were 34.1, 16.7, 15.5, 7.2, and 7.0 months, respectively. The mPFS for the overall cohort was 2.2 months, and the mOS was 7.6 months [40].

### 3.4. Pooled Analyses of ICIs’ Efficacy in VSCC

A pooled analysis of ORR was conducted, including all six single-arm trials encompassing 181 patients. Reported ORRs ranged from 5.6% to 75.0%. Using a random-effects model, the pooled ORR was estimated at 21.0% (95% CI: 6.0–53.0%) (Figure 2). However, high heterogeneity was observed across studies (I^2^ = 85.6%), reflecting differences in treatment strategies (monotherapy vs. combination), patient selection (PD-L1 status), and study size.

Analyzing the three studies of single-agent ICI therapy (KEYNOTE-158, KEYNOTE-028, CheckMate 358; n = 124), the pooled ORR was 11% (95% CI: 6–18%) with no heterogeneity (I^2^ = 0%) [24,25,37] (Figure 3).

A subgroup analysis comparing single-agent versus combination regimens (Yeku et al., SWOG S1609; n = 40) found a higher ORR with combination therapy (46%, 95% CI: 0–100%, I^2^ = 90.4%) versus monotherapy (11%, 95% CI: 5–22%, I^2^ = 0%), although the difference did not reach statistical significance (*p* = 0.13), indicating a possible but not statistically confirmed difference in efficacy between single-agent and combination strategies [39,40] (Figure 4).

An exploratory analysis of PD-L1 expression showed a pooled ORR of 21% (95% CI: 15–29%) among confirmed PD-L1-positive patients (n = 131) and 29% (95% CI: 4–71%) among the small subgroup of known PD-L1-negative patients (n = 7). The combined pooled ORR for both groups was 22% (95% CI: 16–29%) using a fixed-effect model with no heterogeneity (*p* = 0.6546) (Figure 5 and Figure 6).

In a separate analysis, studies were stratified by PD-L1-positive versus PD-L1-unknown/unselected populations, with pooled ORRs of 22% (95% CI: 4–64%) and 20% (95% CI: 11–35%), respectively. No significant subgroup difference was observed (*p* = 0.9173) (Figure 7).

A pooled analysis of mPFS was conducted across four single-arm clinical trials [24,37,38,40]. The reported mPFS values ranged from 1.3 to 2.5 months. The pooled mPFS, estimated using a random-effects model, was 2.2 months (95% CI: 1.72–2.74) with substantial heterogeneity (I^2^ = 70.2%), likely due to variation in treatment regimens (monotherapy vs. combination), treatment lines, and patient characteristics (Figure 8). These findings highlight the limited disease control offered by current ICI strategies and support the need for biomarker-enriched or combination approaches.

A meta-analysis of mOS from the same four studies revealed a pooled mOS of 6.4 months (95% CI: 0.68–10.16) using a random-effects model [24,27,38,40]. The reported median OS values ranged from 3.7 to 17.5 months, with moderate heterogeneity (I^2^ = 58.3%) reflecting clinical variability in therapeutic approach, prior treatments, and patient selection (Figure 9).

### 3.5. Safety Outcomes and Meta Analysis

In KEYNOTE-158, 50.5% of patients experienced treatment-related AEs, with 11.9% experiencing ≥G3 AEs, including two treatment-related deaths [37]. KEYNOTE-028 reported 66% of patients experiencing AEs, with 14% experiencing ≥G3 toxicities [24]. The PEVOsq trial had 23.5% of patients experiencing ≥G3 AEs [38]. CheckMate 358 reported 100% of patients experiencing G1–2 AEs, but no ≥G3 events [25]. In the study by Yeku et al., any-grade AEs occurred in 83% of patients, with 45.8% experiencing grade ≥ 3 events, though no treatment-related deaths were reported [39]. In SWOG S1609, 31.3% of patients experienced AEs, including 25% with grade ≥ 3 toxicity and one treatment-related death (6.7%) [40]. A pooled safety analysis was conducted. The incidence of any-grade AEs, reported in three studies that included 130 patients, ranged from 50% to 100%, with a pooled estimate of 73% (95% CI: 13–98%) using a random-effects model [25,37,39]. Substantial heterogeneity was observed (I^2^ = 79.2%, *p* = 0.0082), likely due to variability in treatment regimens and reporting standards (Appendix A). Grade ≥ 3 AEs were assessed in five studies involving 163 patients, with a pooled incidence of 23% (95% CI: 10–45%) and moderate-to-substantial heterogeneity (I^2^ = 70.6%, *p* = 0.0087) [25,37,38,39,40]. These high-grade toxicities, although infrequent, highlight the need for close safety monitoring (Appendix A). Finally, treatment-related deaths (grade 5 AEs) were reported in five studies, including 157 patients, with a pooled incidence of 3% (95% CI: 1–8%) under a fixed-effect model and no observed heterogeneity (I^2^ = 0%, *p* = 0.8379) [24,25,37,38,40] (Appendix A).

## 4. Discussion

VSCC remains a rare but challenging gynecologic malignancy, particularly in its advanced and recurrent forms, where treatment options are limited and prognosis is poor. While early-stage disease can often be managed with surgical resection and adjuvant therapies, patients with metastatic or refractory VSCC face dismal survival outcomes, with a five-year survival rate as low as 14% in distant recurrences [3,4,5]. Given the increasing role of immunotherapy in HPV-associated cancers and other solid tumors, ICIs have emerged as a promising therapeutic option for select patients with VSCC [24,25,37,38,39,40]. However, due to the rarity of the disease, clinical evidence supporting the use of ICIs remains scarce, and treatment decisions are often extrapolated from studies in cervical and head and neck cancers [13,14,15]. This systematic review and meta-analysis evaluated ICIs in advanced VSCC specifically: ORR was encouraging, despite a short PFS and a modest mOS (6–7 months). ICIs were generally well-tolerated, as no unexpected toxicities emerged, and the benefit tended to be durable (mDoR ~ 20 months in responders). A recent systematic review and single-arm meta-analysis assessed the efficacy of pembrolizumab in advanced vulvar cancer [41]. Their findings indicated an ORR of 10.9%, with a median OS of 6.2 months and a median PFS of 2.1 months, consistent with our analysis [41]. The PEVOsq trial evaluated pembrolizumab with vorinostat and reported the longest OS (17.5 months), suggesting that combination strategies may enhance long-term survival outcomes [38]. In fact, our subgroup analyses revealed that, while monotherapy showed modest responses, combination strategies achieved notably longer OS, suggesting that synergy with epigenetic modulators may play a critical role in immune responsiveness in VSCC. Additionally, the observed durability of response (mDoR ~ 20 months) indicates that even a small proportion of responders may derive long-term benefit, reinforcing the need for precise patient selection. Further studies to determine whether dual or multiple therapy strategies could improve outcomes in advanced VSCC are warranted. Interestingly, the KEYNOTE-158 trial, which included 101 VSCC patients, demonstrated no clear correlation between PD-L1 expression and response, as PD-L1-negative tumors exhibited a higher ORR (28.6%) than PD-L1-positive tumors (9.5%) [37]. This contrasts with the general evidence and data from cervical cancer, where PD-L1 positivity is often predictive of response to pembrolizumab [15]. In our subgroup analysis, the response rates did not correlate with PD-L1 status, echoing a similar uncertainty observed in other HPV-related malignancies, where PD-L1 is not always a reliable biomarker [13,42]. A recent meta-analysis examined the positivity rate of PD-L1 expression and its clinical significance in vulvar cancer and notably found that PD-L1 expression was associated with worse OS and PFS [43]. These findings suggest that, while PD-L1 is frequently expressed in VSCC and may serve as a prognostic biomarker, its role as a predictive marker for response to ICIs remains uncertain in this malignancy. This also highlights a potential heterogeneity in PD-L1 scoring methodologies across studies, in terms of evaluation methodology, such as the analysis of biopsies that do not always reflect the molecular status of the entire tumor, and suggests that alternative PD-L1 thresholds or evaluation methods may be needed to optimize patient selection. Notably, an analysis of PD-L1 expression in 84 vSCC patients found that PD-L1 positivity in peritumoral immune cells, but not in cancer cells, was independently associated with improved overall survival [44]. This study underscores the importance of evaluating PD-L1 expression in both tumor and immune compartments and supports the notion that PD-L1-positive immune microenvironments may reflect an active anti-tumor response. The CheckMate 358 trial, which assessed nivolumab in HPV-positive VSCC, reported an ORR of 20%, suggesting that HPV-driven tumors may exhibit greater immune responsiveness, a hypothesis that is consistent with the findings in other HPV-related cancers, such as oropharyngeal and anal carcinoma [25]. Most current studies for advanced VSCC, however, lack HPV status data, and subgroup meta-analyses for HPV status are, thus, not feasible. However, HPV positivity, generally considered a favorable prognostic and predictive factor in other cancers, has not yet demonstrated a consistent predictive role in VSCC, possibly due to the histological and immunological heterogeneity of HPV-driven versus HPV-independent tumors [45,46]. Given the discrepancy between trials, future research should focus on refining PD-L1 cutoffs and assessment techniques and exploring additional predictive biomarkers, such as TMB, MSI status, and TILs, to enhance patient stratification. In comparison to the broader ICI literature, the response rates observed in VSCC appear lower than those reported in other HPV-associated malignancies. For instance, PD-1 blockade in recurrent or metastatic cervical cancer has yielded ORRs ranging from 15% to 25% in biomarker-selected populations [13]. Similarly, checkpoint inhibition in anal and oropharyngeal squamous carcinomas has shown encouraging results in both monotherapy and combination settings [14,47]. This suggests that, while VSCC may share certain immunobiological features with these tumors, it may also harbor distinct resistance mechanisms (i.e., stromal barriers, immune exclusion, or epigenetic repression of immune signaling) that require further investigation Additionally, the limited number of clinical trials and small sample sizes in VSCC highlight the necessity of collaborative efforts to establish larger, multicenter prospective studies to better characterize the role of ICIs in this disease. To our knowledge, the first meta-analysis to comprehensively evaluate the efficacy and safety of ICIs, both alone and in combination, in advanced VSCC, offering pooled estimates derived from multiple trials. Our strict inclusion criteria, dual independent screening, and subgroup analyses enhance the reliability and completeness of our findings. Nevertheless, some limitations must be acknowledged. First, the heterogeneity of the included studies poses a challenge for direct comparisons. The trials varied not only in terms of prior treatment exposure and PD-L1 assessment methods but also regarding the type of immunotherapy regimen employed. Specifically, some studies evaluated ICIs as monotherapy, while others combined them with chemotherapy and/or radiotherapy. This variation may have influenced treatment outcomes and complicated direct comparisons, as the response rates in combination regimens tend to be higher and may not reflect the efficacy of immunotherapy alone in routine clinical settings.

Additionally, while our meta-analysis pooled data from multiple studies, the small sample sizes may limit the generalizability of our findings. Second, there is a lack of long-term follow-up data for many included studies, particularly regarding the durability of response and late-onset immune-related AEs. Furthermore, our study is limited by the absence of RCTs, limiting the ability to compare ICIs with standard-of-care treatments. Given the rarity of VSCC, most treatment recommendations are extrapolated from cervical and other HPV-associated cancers, and the current evidence on ICIs is derived from basket trials or single-arm studies. This underscores the urgent need for larger, well-powered prospective studies to establish the role of immunotherapy in VSCC and define optimal treatment strategies. Despite these limitations, our study provides valuable insights into the potential role of ICIs in a challenging setting as advanced VSCC, offering a foundation for future research as well as broadly accepted, VSCC-specific clinical guidelines.

## 5. Conclusions

This systematic review and meta-analysis demonstrate that ICIs provide a promising and well-tolerated therapeutic option for patients with advanced VSCC. The pooled mPFS and mOS reflect the limited, yet clinically relevant, efficacy of these agents for this complicated clinical setting. However, response variability underscores the need for improved patient stratification, as PD-L1 expression or HPV status were unreliable predictors of response, thus reinforcing the necessity of additional biomarkers to optimize patient selection.

## Figures and Tables

**Figure 1 cancers-17-02392-f001:**
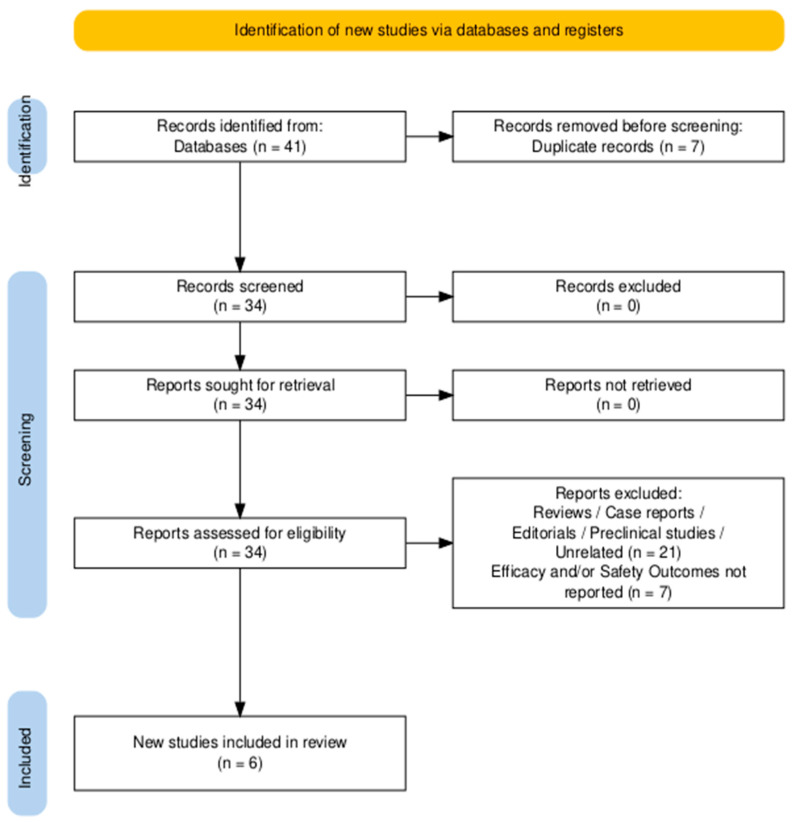
PRISMA flowchart diagram of studies’ selection.

**Figure 2 cancers-17-02392-f002:**
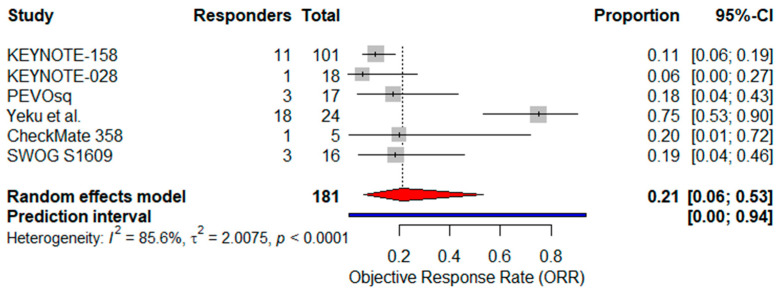
Forest plot showing the pooled objective response rate (ORR) across six studies investigating immunotherapy in advanced VSCC. The pooled ORR was estimated using a random-effects model. The red diamond indicates the pooled effect size; the blue line represents the 95% prediction interval [24,25,37,38,39,40].

**Figure 3 cancers-17-02392-f003:**
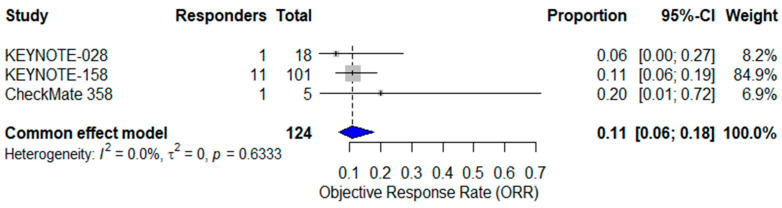
Forest plot of objective response rates (ORR) from three studies evaluating single-agent immune checkpoint inhibitors in advanced vulvar squamous cell carcinoma (VSCC). The pooled ORR, calculated using a common-effect (fixed-effect) model, was 11% (95% CI: 6–18%). No heterogeneity was observed (I^2^ = 0%). The blue diamond represents the pooled effect estimate [24,25,37].

**Figure 4 cancers-17-02392-f004:**
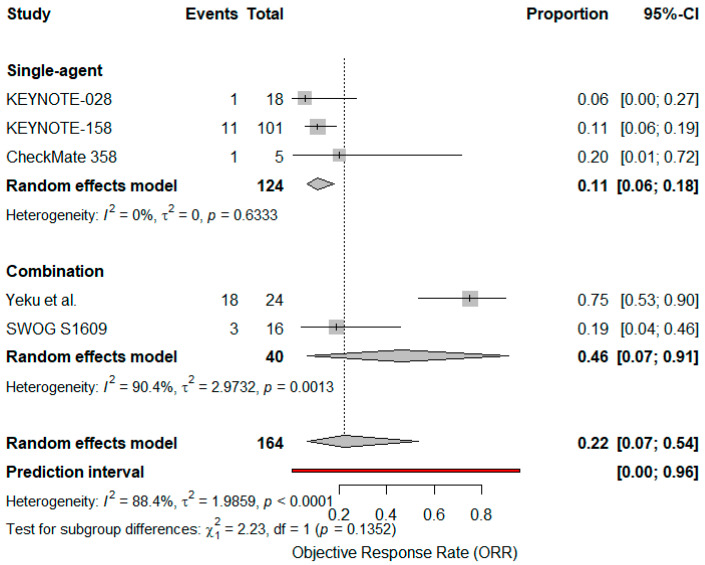
Forest plot comparing objective response rates (ORR) for single agent vs. combination immunotherapy in advanced VSCC. Diamonds represent pooled effect estimates for each subgroup and overall; the blue line indicates the prediction interval [24,25,37,39,40].

**Figure 5 cancers-17-02392-f005:**
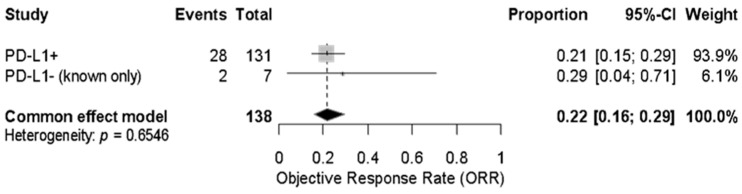
Forest plot showing the objective response rate (ORR) to immune checkpoint inhibitors in patients with advanced vulvar squamous cell carcinoma, stratified by confirmed PD-L1 status. The pooled ORR across both PD-L1-positive (n = 131) and PD-L1-negative (n = 7) patients was 22% (95% CI: 16–29%) under a fixed-effect model. No significant heterogeneity was observed (*p* = 0.6546).

**Figure 6 cancers-17-02392-f006:**
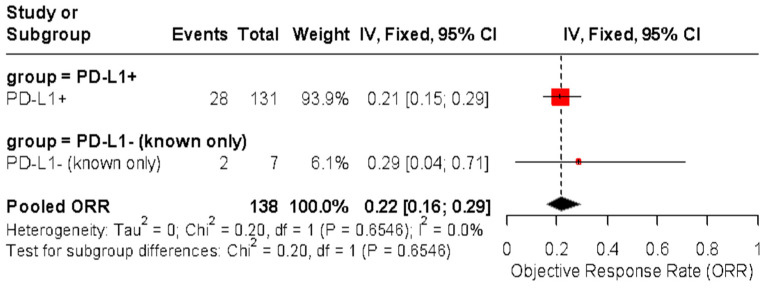
Forest plot of objective response rate (ORR) in patients with advanced vulvar squamous cell carcinoma treated with immune checkpoint inhibitors, stratified by confirmed PD-L1 status. The pooled ORR was 21% (95% CI: 15–29%) in PD-L1-positive tumors and 29% (95% CI: 4–71%) in PD-L1-negative tumors. No significant difference was observed between subgroups (*p* = 0.6546, I^2^ = 0%).

**Figure 7 cancers-17-02392-f007:**
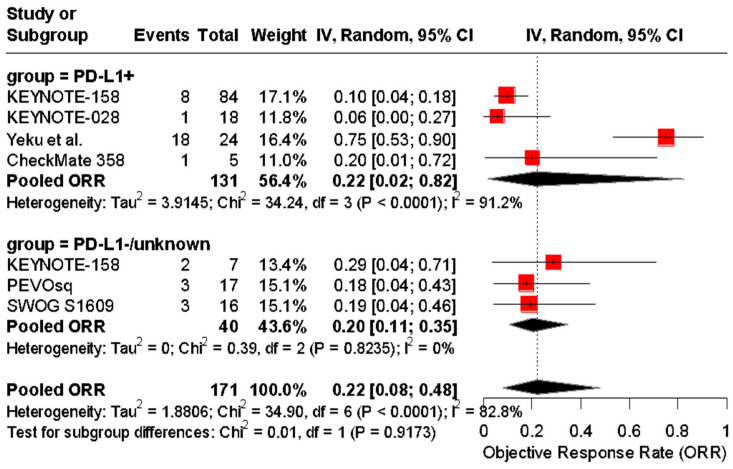
Forest plot of objective response rates (ORRs) to immune checkpoint inhibitors in patients with advanced vulvar squamous cell carcinoma, stratified by PD-L1 status. The pooled ORR was 22% (95% CI: 4–64%) in PD-L1-positive patients and 20% (95% CI: 11–35%) in PD-L1-negative or untested patients. No significant subgroup difference was detected (Chi^2^ = 0.01, *p* = 0.9173) [24,25,37,38,39,40].

**Figure 8 cancers-17-02392-f008:**
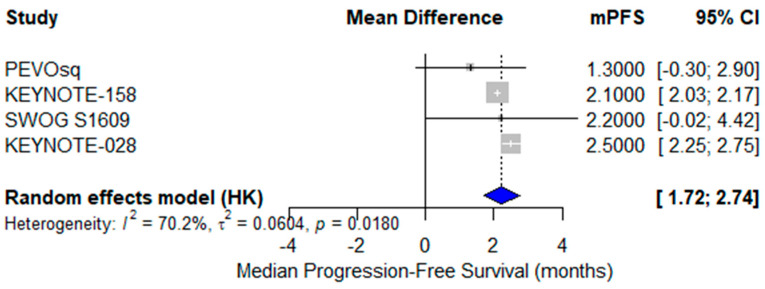
Forest plot of median progression-free survival (mPFS) across four studies evaluating immunotherapy in advanced vulvar squamous cell carcinoma (VSCC). The pooled mPFS was 2.2 months (95% CI: 1.72–2.74), calculated using a random-effects model. Moderate to high heterogeneity was observed (I^2^ = 70.2%).

**Figure 9 cancers-17-02392-f009:**
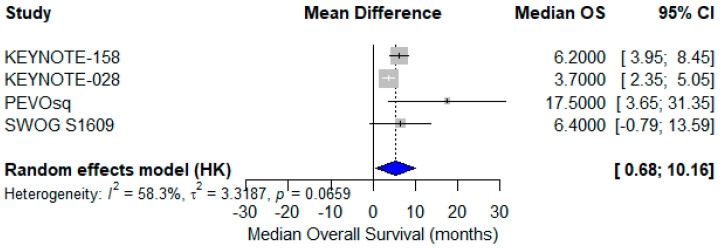
Forest plot showing median overall survival (mOS) across four studies evaluating immune checkpoint inhibitors in advanced vulvar squamous cell carcinoma (VSCC). The pooled mOS was 6.4 months (95% CI: 0.68–10.16), calculated using a random-effects model. Moderate heterogeneity was observed (I^2^ = 58.3%) [24,37,38,40].

**Table 1 cancers-17-02392-t001:** Summary of the included studies.

Study	Phase/Design	Population	PD-L1	Intervention	Efficacy	Safety	Key Findings
KEYNOTE-158 [37]	II, single arm	101 advanced (metastatic, unresectable) VSCC Median age 64 (31–87) 88.1% stage IV 19.8% M0 II 3% III 10%	84 PD-L1+ 7 PD-L1− 10 unknown	Pembrolizumab (200 mg IV q3w up to 35 cycles or 2 yrs)	ORR 10.9% (1 CR 10 PR) PD-L1+: ORR 9.5% PD-L1−: ORR 28.6% mPFS 2.1 months mOS 6.2 months mDoR 20.4 months	50.5% AEs 11.9% G3–5 (2 deaths)	Durable responses in a subset of patients regardless of PD-L1 status.
KEYNOTE-028 [24]	Ib, single arm	18 advanced (metastatic, unresectable) VSCC	PD-L1+	Pembrolizumab (10 mg/kg q2w for up to 2 yrs)	ORR 5.6% mPFS 2.5 months mOS 3.7 months mDoR 3.9 months	66% AEs 14% ≥ G3 4% TD	Higher PD-L1 expression correlated with better ORR and PFS
PEVOsq basket trial [38]	II, single arm	17 advanced VSCC Median age 63 (40–85) 14 metastatic 3 relapsed (unresectable)	NA	Pembrolizumab (200 mg q3w IV) with vorinostat (400 mg QD PO) FIRST LINE	ORR 18.8% (1 CR and 2 PR) mPFS 1.3 months mOS 17.5 months	23.5% G3/4 AEs	Encouraging efficacy in VSCC
Yeku et al. [39]	II, single arm	24 advanced VSCC 22 unresectable 2 recurrent	PD-L1+	Pembrolizumab (200 mg q3w IV) with cisplatin and RT	ORR 75% 6-month RFS rate 70% mPFS NR	78.6% G3/4 AEs	Combined pembrolizumab improved ORR and 6-month RFS
CheckMate 358 [25]	I/II, single arm	5 advanced (locally, metastatic) VSCC Median age 59 (40–78) 1 IIB 1 IIIB/C 3 IVA/B 2 HPV+ 3 HPV−	PD-L1+	Nivolumab (240 mg IV q2w) 1 first line 4 one or 2 prior lines (Platinum)	ORR 20% (1 PR, 3 SD) DCR 80% mDoR 5 months 6-month PFS rate 40.0% 12-month OS rate 40% 18-month OS rate 20.0%	100% G1–2 AEs 0% G3–5	Encouraging responses from Nivolumab monotherapy, additional investigation warranted
SWOG S1609 trial [40]	II, single arm	16 advanced VSCC Median age 55.5	NA	Nivolumab (240 mg IV q2w) with ipilimumab (1 mg/kg IV q6w)	ORR 18.8% mPFS 2.2 months mOS 7.6 months	G1/2 25% G3/4 25% G5 6.7% (1 death)	Objective responses lasting over 1 year

AE(s): adverse event(s); CR: complete response; DCR: disease control rate; DoR: duration of response; G: grade; IV: intravenous; NA: not available; NR: not reached; ORR: objective response rate; OS: overall survival; PD-L1: programmed death ligand 1; PFS: progression-free survival; PR: partial response; q(n)w: every (n) weeks; RFS: recurrence-free survival; RT: radiotherapy; TD: treatment discontinuation; VSCC: vulvar squamous cell carcinoma; yrs: years.

## Data Availability

The authors confirm that the data supporting the findings of this study are available within the article and/or its Appendix A.

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
