# Peer review of "Immunotherapy and Advanced Vulvar Cancer: A Systematic Review and Meta-Analysis of Survival and Safety Outcomes"

_cancers, 2025, doi:10.3390/cancers17142392_

Round 1
Reviewer 1 Report (Previous Reviewer 3)
Comments and Suggestions for Authors
Thank you for this much improved manuscript.
My main concern is that data form various study designs have been taken together. The ORR of a single-arm IO study will be expected much lower than that of a study combining IO with radiotherapy and chemotherapy. This has not been addressed in the results. This results in a more optimistic outcome of the benefits of IO than may we true in clinical practice.
Author Response
Comment 1: My main concern is that data form various study designs have been taken together. The ORR of a single-arm IO study will be expected much lower than that of a study combining IO with radiotherapy and chemotherapy. This has not been addressed in the results. This results in a more optimistic outcome of the benefits of IO than may we true in clinical practice.
Response 1: We thank the Reviewer for this important observation and for the invaluable time spent reviewing our work. We would like to clarify that the Results section (Subgroup analysis) explicitly presents ORR estimates separately for single-agent ICI studies and those using combination regimens. We observed that combination regimens were associated with higher ORR (46%) compared to monotherapy (11%), although the difference did not reach statistical significance. This stratification is visualized in Figure 4. To further address your concern, we have also revised the limitations section in the Discussion paragraph to acknowledge that this heterogeneity in treatment approach may contribute to an overestimation of benefit when pooling data. We now explicitly state that combination regimens may inflate response rates.
Reviewer 2 Report (Previous Reviewer 5)
Comments and Suggestions for Authors
This manuscript is an improved version of the previous one.
Neverthless, this is not an original research, this is a systematic review and it should be defined correctly.
Please follow the requirements of the journal. It is important for other healthcare proffesionals to better and faster identify, by key words in data bases, this review of literature.
Author Response
We would like to thank the reviewer for the time spent revising our work.
Comment 1: Neverthless, this is not an original research, this is a systematic review and it should be defined correctly. Please follow the requirements of the journal. It is important for other healthcare proffesionals to better and faster identify, by key words in data bases, this review of literature.
Response 1: Thank you. After discussing with the editors, our article has been redefined as "Systematic Review".
Round 2
Reviewer 1 Report (Previous Reviewer 3)
Comments and Suggestions for Authors
I am gratefull for the changes made to the manuscript. I still think that the conclusion is too optimistic: ICIs provide a meaningful and well-tolerated therapeutic option for patients with advanced VSCC. Thereafter, the messag is put into more perspective, but opening with it sets the standard.
Author Response
Thank you again for your time.
Comment 1: I still think that the conclusion is too optimistic: ICIs provide a meaningful and well-tolerated therapeutic option for patients with advanced VSCC. Thereafter, the messag is put into more perspective, but opening with it sets the standard.
Response 1: Thank you. We revised the statement and the conclusions accordingly to give a more nuanced and realistic view.
This manuscript is a resubmission of an earlier submission. The following is a list of the peer review reports and author responses from that submission.
Round 1
Reviewer 1 Report
Comments and Suggestions for Authors
Dear Authors,
I read your manuscript "Immunotherapy and Vulvar Cancer: A Systematic Review and Meta-Analysis of Survival Outcomes". It is very interesting and well written article. All the metrics are provided along with figures. However I have one comment.
Given the magnitude of differences observed between 3 studies in fig 3 and 2 studies in fig 4, do you think it is any conclusive ? How would you interpret the result ?
Best.
Reviewer 2 Report
Comments and Suggestions for Authors
Several issues have been identified in this manuscript. I have grouped them according to the manuscript’s structure, as outlined below:
Background
The authors state that several biomarkers — including PD-L1 expression, microsatellite instability (MSI)/deficient mismatch repair (dMMR), tumor mutational burden (TMB), and the presence of tumor-infiltrating lymphocytes (TILs) — have been identified as potential predictors of response to immune checkpoint inhibitors (ICIs) [10-13]. However, none of these references specifically address vulvar squamous cell carcinoma (vSCC).
The manuscript lacks an analysis of the scientific rationale for the use of immunotherapy in vSCC. Although several studies have demonstrated the predictive value of TILs in vSCC, they also highlight different immune cell signatures between HPV-positive and HPV-negative cases.
Notably, an important study (not cited by the authors) demonstrated that prognosis in vSCC depends on the distribution of PD-L1 expression between cancer cells and stromal immune cells, suggesting distinct roles in therapeutic response — a critical finding, particularly before the clinical introduction of PD-L1-targeting ICIs. The study found that PD-L1 positivity in peritumoral immune cells was an independent favorable prognostic factor for overall survival. These results highlight the importance of a comprehensive assessment of PD-L1 expression in both tumor and immune cells. Moreover, PD-L1 expression on peritumoral immune cells may serve as an additional prognostic biomarker and could influence the effectiveness of anti-PD-L1 therapies in vSCC patients.
These preclinical insights were partially supported by the KEYNOTE-158 trial, where the objective response rate (ORR) was 10.9% (95% CI: 5.6–18.7%). Interestingly, PD-L1-positive tumors achieved an ORR of 9.5%, whereas PD-L1-negative tumors showed a higher ORR of 28.6%. This unexpected result may be partly explained by the methods used to assess PD-L1 expression, which often relied on tissue microarrays rather than full-face tumor sections, potentially leading to sampling bias and inaccurate representation of PD-L1 status.
Additionally, preclinical studies suggest that future trials should stratify HPV-dependent and HPV-independent vSCC cases separately, given their distinct prognoses, primarily influenced by differences in radiotherapy response. Most patients included in the reviewed studies received radiotherapy, which could have affected the combined therapeutic outcomes and further underscores the need for stratification. However, the authors did not address this critical consideration. Moreover, they failed to acknowledge that, despite basic science raising doubts about the efficacy of ICIs in vSCC, clinicians nonetheless proceeded with clinical investigations — primarily through single-arm studies assessing objective response rates — and case reports..
The authors also claim that “Recent studies have provided initial evidence supporting the use of ICIs in VSCC [17],” but reference [17] is merely a review article — not primary clinical evidence — and therefore carries a low level of evidence.
Issues with References
There is a significant misuse of references, indicating a lack of rigor. For example, the background states:
"The U.S. Food and Drug Administration (FDA) has approved pembrolizumab for any solid tumor with PD-L1 positivity, TMB-high, or MSI-high/dMMR status, and the National Comprehensive Cancer Network (NCCN) guidelines recognize pembrolizumab as a second-line option for advanced or metastatic VSCC meeting these biomarker criteria [18-20]."
However, the cited references are inappropriate:
• [18] is a summary of FDA approval for TMB-high tumors but does not specifically address VSCC.
• [19] and [20] are manuals for statistical software (IBM SPSS and R), completely unrelated to the clinical context.
Moreover, these same references are also incorrectly cited to support the statement that due to the rarity of VSCC, treatment decisions are extrapolated from other cancer types. This inconsistent referencing undermines the credibility of the manuscript.
Materials and Methods
The search strategy was poorly designed. Although the keywords were appropriate, the search should have been strictly limited to clinical trials (RCTs/non-RCTs) and meta-analyses in English. Instead, their broad search strategy led to the identification of 104 studies, of which 95 were excluded, and only 9 were included. This unnecessary broad search gives a false impression of extensive effort but reflects poor planning.
Case reports and retrospective cohorts should have been excluded from a systematic review aiming for higher levels of evidence.
Results
There is significant heterogeneity across the included studies: unknown disease stages (advanced/metastatic stages unclear), mixed HPV status, varying ICI agents (PD-1 vs. PD-L1 inhibitors), and different therapeutic combinations (ICI alone vs. ICI plus other agents).
Median age also varied widely (40 to 88 years), making comparisons difficult.
Consequently, across the nine studies analyzed, the ORR ranged from 6% to 100%. Clinical trials specifically reported ORRs between 6% and 20%, and the median overall survival (mOS) varied from 3.8 to 17.5 months.
These wide-ranging results further support that this systematic review — primarily based on four single-arm clinical trials, one retrospective cohort, and three case reports — is not clinically conclusive.
Discussion
The discussion largely repeats the background and results without providing new insights or proper interpretation. References are again misused.
Reviewer 3 Report
Comments and Suggestions for Authors
The authors provide a systematic review and meta-analysis of survival outcomes of ICI in patients with vulvar cancer.
- Case reports have been included, stating ORR of 100%. Obviously this is publication bias. I would advice to dismiss the case reports.
- Recently, a study combining anti-CTLA4 and a PD-1 blocker with 16 patients with vulvar cancer has been published (Clin Cancer Res 2025; 31: 308-15. It would provide new interesting data.
- The study with most patients (Keynote-158) comprises of 2/3 of all patients in this analysis. The results of this study are poor with regard to efficacy. So, the conclusion of this analysis is inappropriate. An ORR of only 14% with a pooled mean OS of 6,2 months is not worth to pursue from a societal benefit as well as an individual benefit/harm perspective (25% grade 3 side effects).
Reviewer 4 Report
Comments and Suggestions for Authors
Dear Authors!
Although the numbers of eligible studies and patients included are low, this meta-anaylsis for advanced vulvar carcinoma treated with immune checkpoint inhibitors is very important.
Please address a few points:
Integration of case reports into a meta-analysis is prone to induce a bias. The described prerequisites for inclusion of studies are not met by case reports. So, please make sure form the beginning on that they are only treated as background context.
L.181: Four case reports are mentioned, beforehand only three (L.167 and abstract), L336
Insert reference numbers in Fig. 2 to enhance clarity.
Fig. 2: please adapt legend for bottom axis (either % numbers or decimal fractions
3.4 Please add that pooled ORR findings are not significant while pooled PFS results are.
From the data you can draw from the studies included in your meta-analysis, can you provide pooled calculations for HPV-positive vs. HPV-negative cases? That would add important information to your work. Include this topic also in the Discussion, L. 342.
Were the four case report patients HPV-positive?
356ff. try to find a biological explanation for the paradox findings of the ORR in PD-L1-positive / -negative patients. Which PD-L1 scoring systems were the results based on? Comparable to Meister et al.?
388ff. Also they vary in the monoclonal antibody as an active ingredient oft he applied drugs.
Reviewer 5 Report
Comments and Suggestions for Authors
The authors presented a systematic review and meta-analysis of Immunotherapy in Vulvar Cancer.
They defined their manuscript as article, probably based on relatively few numbers of articles. This is neither an article (original research) nor a review.
I used “Immunotherapy in Vulvar Cancer” in advanced search in Pubmed and I received 116 articles in the last 5 years.
I used ((immunotherapy) AND (advanced or metastatic)) AND (vulvar cancer) as search items and I found 51 articles in the last 5 years.
My point is that the methodology has major flaws. So, the results can’t be accurately interpreted.
I consider that this manuscript has major flaws and need major work.